# Non-invasive quantification of the mitochondrial redox state in livers during machine perfusion

Reinier J. de Vries[1,2,3], Stephanie E. J. Cronin[1,2], Padraic Romfh[4], Casie A. Pendexter[1,2], Rohil Jain[1,2], Benjamin T. Wilks[1,2], Siavash Raigani[1,2,5], Thomas M. van Gulik[3], Peili Chen[4], Heidi Yeh[5], Korkut Uygun[1,2], Shannon N. Tessier[1,2]*

1 Center for Engineering in Medicine and Surgery, Harvard Medical School and Massachusetts General Hospital, Boston, MA, United States of America, 2 Shriners Hospitals for Children—Boston, Boston, MA, United States of America, 3 Department of Surgery, Amsterdam University Medical Centers–Location AMC, University of Amsterdam, Amsterdam, the Netherlands, 4 Pendar Technologies, Cambridge, MA, United States of America, 5 Division of Transplantation, Department of Surgery, Massachusetts General Hospital, Boston, MA, United States of America

* sntessier@mgh.harvard.edu

**Data Availability Statement:** The authors declare that the data supporting the findings of this study are available within the paper and its

## Abstract

Ischemia reperfusion injury (IRI) is a critical problem in liver transplantation that can lead to life-threatening complications and substantially limit the utilization of livers for transplantation. However, because there are no early diagnostics available, fulminant injury may only become evident post-transplant. Mitochondria play a central role in IRI and are an ideal diagnostic target. During ischemia, changes in the mitochondrial redox state form the first link in the chain of events that lead to IRI. In this study we used resonance Raman spectroscopy to provide a rapid, non-invasive, and label-free diagnostic for quantification of the hepatic mitochondrial redox status. We show this diagnostic can be used to significantly distinguish transplantable versus non-transplantable ischemically injured rat livers during oxygenated machine perfusion and demonstrate spatial differences in the response of mitochondrial redox to ischemia reperfusion. This novel diagnostic may be used in the future to predict the viability of human livers for transplantation and as a tool to better understand the mechanisms of hepatic IRI.

## Introduction

Ischemia reperfusion injury (IRI) is a major complication of liver surgery and can provoke not just liver, but multiorgan failure [1, 2]. Because there are currently no early diagnostics available to predict the extent of post-reperfusion injury, IRI is a particular challenge in liver transplantation that can result in early allograft dysfunction, primary nonfunction, chronic and acute rejection [3, 4]. Additionally, fear of IRI directly contributes to the global donor organ shortage, as the lack of pre-implantation diagnostics also leads to usable organs being discarded empirically [5]. While the total number of liver donations after brain death (DBD) and donations after circulatory death (DCD) have increased over the years, about 7.1% of DBD livers and 30% of DCD livers are discarded. As many as 58.5% of DCD discards meet the

Supplementary Information files. Any additional data, if needed, will be provided upon request.

**Funding:** This research was funded from the US National Institutes of Health (R01DK114506, R01DK096075 to KU; R01DK107875 and R43DK120127-01 to KU and SNT; R01HL157803 to SNT). This material is partially based upon work supported by the National Science Foundation under Grant No. EEC 1941543 (to KU and SNT). Further, we gratefully acknowledge funding to SNT for Career Development from NIH (K99/R00HL143149), American Heart Association (18CDA34110049), Harvard Medical School Eleanor and Miles Shore Fellowship, and the Claflin Distinguished Scholar Award on behalf of the MGH Executive Committee on Research (ECOR). We thankfully acknowledge support provided by the Tosteson Fellowship awarded to RJV by the MGH ECOR. We also acknowledge the Interim Support Funding to KU by MGH ECOR. Pendar provided support in the form of salaries for authors P.R. and P.C. but did not have any additional role in the study design, data collection and analysis, decision to publish, or preparation of the manuscript. The specific roles of these authors are articulated in the 'author contributions' section.

**Competing interests:** The authors declare competing interests. Drs. de Vries, Uygun, Tessier, and Chen, and Mr. Romfh have provisional patent applications relevant to this study. P.R. and P.C. are employees and shareholders of Pendar Technologies. All competing interests are managed by the MGH and Partners HealthCare in accordance with their conflict-of-interest policies, except for P. R. and P.C. who are subject to the Research Integrity Policy of Pendar Technologies. The following patented technologies have been used in this study: US 7,113,814 Tissue Interrogation spectroscopy (fully licensed by Pendar from VCU), and US 2020/0281474 A1 In-vivo monitoring of cellular energetics with Raman spectroscopy (application). Additional patent applications for use in ophthalmology and tissue viability using Resonance Raman Spectroscopy have been submitted. This does not alter our adherence to PLOS ONE policies on sharing data and materials. All other authors do not have competing interests.

standard acceptance criteria proposed by American Society of Transplant Surgeons [6]. Many waitlisted patients die or become too sick to undergo transplantation, representing several thousand deaths a year in the US alone [7, 8]. For every 4 patients that are removed from the transplant waiting list due to death or illness, 3 unused livers are discarded [5]. Novel diagnostics to quantitatively predict hepatic IRI hold great promise to improve outcomes after liver transplantation and to alleviate the organ shortage through accurate identification of which livers are suitable and unsuitable for transplantation [9].

Hepatic IRI is a multifactorial process that starts with metabolic adaptations in response to ischemia. Although hypoxia alone can result in cell death, less severe hypoxia can lead to metabolic perturbations that initiate fulminant injury upon the sudden influx of oxygen at reperfusion [10, 11]. Anaerobic metabolism and acidosis resulting from ischemia and oxidative stress during reperfusion are considered the most important factors that initiate cell death programs and activate pathways of innate and adaptive immune responses that cause hepatic injury during IRI [11–13]. Mitochondria play a pivotal role in all these factors and may therefore be an ideal target for IRI diagnostics [2, 14, 15].

During normal oxidative phosphorylation, electrons from NADH and FADH2 (that are ultimately transferred to oxygen at mitochondrial membrane complex IV) fuel the proton pumps to generate the potential over the mitochondrial inner membrane that drives ATP generation at complex V [16, 17]. In the absence of oxygen, complex IV remains in a reduced state, and upstream redox changes with the buildup of electrons along the electron transport chain (ETC) impair the mechanisms for controlling free radical production during normal oxidative phosphorylation. This leads to production of reactive oxygen and nitrogen species (ROS and RNS) at complexes I and III when the electrons get transferred all at once upon reperfusion [18]. Downstream, impaired ATP production causes dysfunction of ATP dependent ion transporters ($Na^+/K^+$-ATPase and $Ca^{2+}$-ATPase) which through cytosolic $Ca^{2+}$ influx eventually leads to opening of the mitochondrial permeability transition pore (MPTP) [19]. The depolarization of the inner mitochondrial membrane then further compromises mitochondrial redox status, aggravates ROS/RNS production and causes the releases of pro cell death factors [20]. Thus, lack of oxygen and the consequent reduced status of mitochondrial complex IV play a central role in IRI.

Cytochromes form active sites within the mitochondrial complexes and their spectroscopic properties depend on the redox status [17, 21]. This makes the ETC conducive to spectroscopic assessment and the redox status may be leveraged as a fast and noninvasive diagnostic to predict ischemia reperfusion injury. Cytochrome a,a3 is of special interest as this forms the active site for oxygen in complex IV which–as described–forms the first link in the chain of events that lead to IRI [22]. Not surprisingly, due to its diagnostic potential, extensive efforts have gone into attempts to quantify the cytochrome a,a3 redox status using absorbance spectroscopy [23–27]. In whole organs, however, overlap of absorption spectra of different molecules, inhomogeneous tissue scattering, and edema have limited this approach to describing trends rather than direct quantification of the redox status [27–30]. Alternatively, absorbance spectroscopy has been used to quantify NAD+/NADH and NADP+/NADPH redox status [31]. However, these mostly reflect the redox status in the cytosol, and do not necessarily correspond to the redox status of the ETC.

In contrast to absorbance spectroscopy, Raman spectroscopy depends on the inelastic scattering of monochromatic light to create a spectral profile [32]. This Raman profile is unique for every molecule and strongly dependent on the redox state. In the special case of resonance Raman (RR) spectroscopy, the wavelength of the light source (i.e. the photon energy) is carefully chosen to overlap with an atomic electron transition (absorption peak) of the molecule of interest which strongly enhances the Raman profile against noise from background molecules

[33]. When cytochromes are excited near the Soret absorption band (400 to 450 nm), the intensity of their RR spectra are amplified up to six orders of magnitude. This facilitates detection of minute cytochrome quantities within the complex molecular environment of an organ.

RR spectroscopy has been used to quantify the mitochondrial redox state of isolated mitochondria, cardiomyocytes, and heart tissue *ex vivo* [34–36]. Recently, a new spectral algorithmic approach was introduced to derive the ratio between the reduced and oxidized state of mitochondrial cytochromes from the complex RR spectrum obtained from whole hearts *in vivo*. This RR reduced mitochondrial ratio (3RMR) significantly outperformed all current measures of cardiac ischemia and function [17].

In this study, we adapted the heart specific 3RMR to quantify the mitochondrial redox state in livers during oxygenated machine perfusion and describe the important adaptations that were required to translate this novel technology. Important benefits of the resulting diagnostic are that the hepatic mitochondrial redox status can be objectively quantified via non-contact and label-free optic readings on the liver surface. The readings are rapid, and results are provided real-time and do not require calibration. Additionally, the device is small and portable to aid clinical use. We show that this diagnostic modality can distinguish transplantable versus non-transplantable ischemic rat livers and demonstrate spatial differences in the response of the mitochondrial redox response to hepatic ischemia reperfusion.

## Results

### Spectroscopic quantification of the mitochondrial redox state in livers

Increasing evidence from clinical trials suggests that machine perfusion prior to transplantation alleviates IRI [37–39]. Additionally, machine perfusion provides a unique controlled platform to study metabolically active whole organs *ex vivo* that we leveraged in this study [9, 40, 41]. We collected the complex RR spectra of rat livers during machine perfusion using a custom-built RR spectroscopy system that provides a fast, non-contact, and label-free measurements of the liver tissue. A 441 nm (within the Soret absorption band) excitation wavelength laser and high-resolution spectrometer are situated in a small footprint device. From this device, the laser light is delivered to and the Raman scattering collected from the liver surface via a 1.5 m fiber optic cable which eases *in* and *ex vivo* use (Fig 1). A multitude of design features are included in the device to provide accurate recordings of the RR spectra and are explained in the Materials and methods.

The spectral algorithm to quantify the 3RMR relies on pre-recorded spectral libraries. These libraries are Raman spectra of pure samples of analytes such as bile, beta carotene, etc. that serve as reference for a regression algorithm that fits these libraries onto an unknown Raman spectrum by varying the relative concentrations (denoted by β) of each analyte. These individual spectrum ($x_1$—$x_5$) and their relative weights ($\beta_1$ – $\beta_5$) add up to generate the best fit curve for the original Raman spectrum (Fig 1, $y_r$ and $y_m$ respectively). The procedure of creation of these libraries is provided in the methods section. The advantage of this approach is that the relative contribution of each library spectrum to the complex spectrum of the organ can be accurately quantified even when it varies substantially between different individuals of a species or between different species. Inclusion of the RR spectra of reduced and oxidized mitochondria in the library therefore enables quantification of the mitochondrial redox status and 3RMR. However, it is essential that the spectra of other molecules that exhibit resonance enhancement by the 441 nm excitation wavelength–and thus substantially contribute to the complex RR spectrum–are also included in the algorithm to create an accurate fit to the complex RR spectrum obtained from the whole organ. This complicates translation of this approach from hearts to livers as these organs contain different molecules that are enhanced at the 441 nm excitation wavelength.

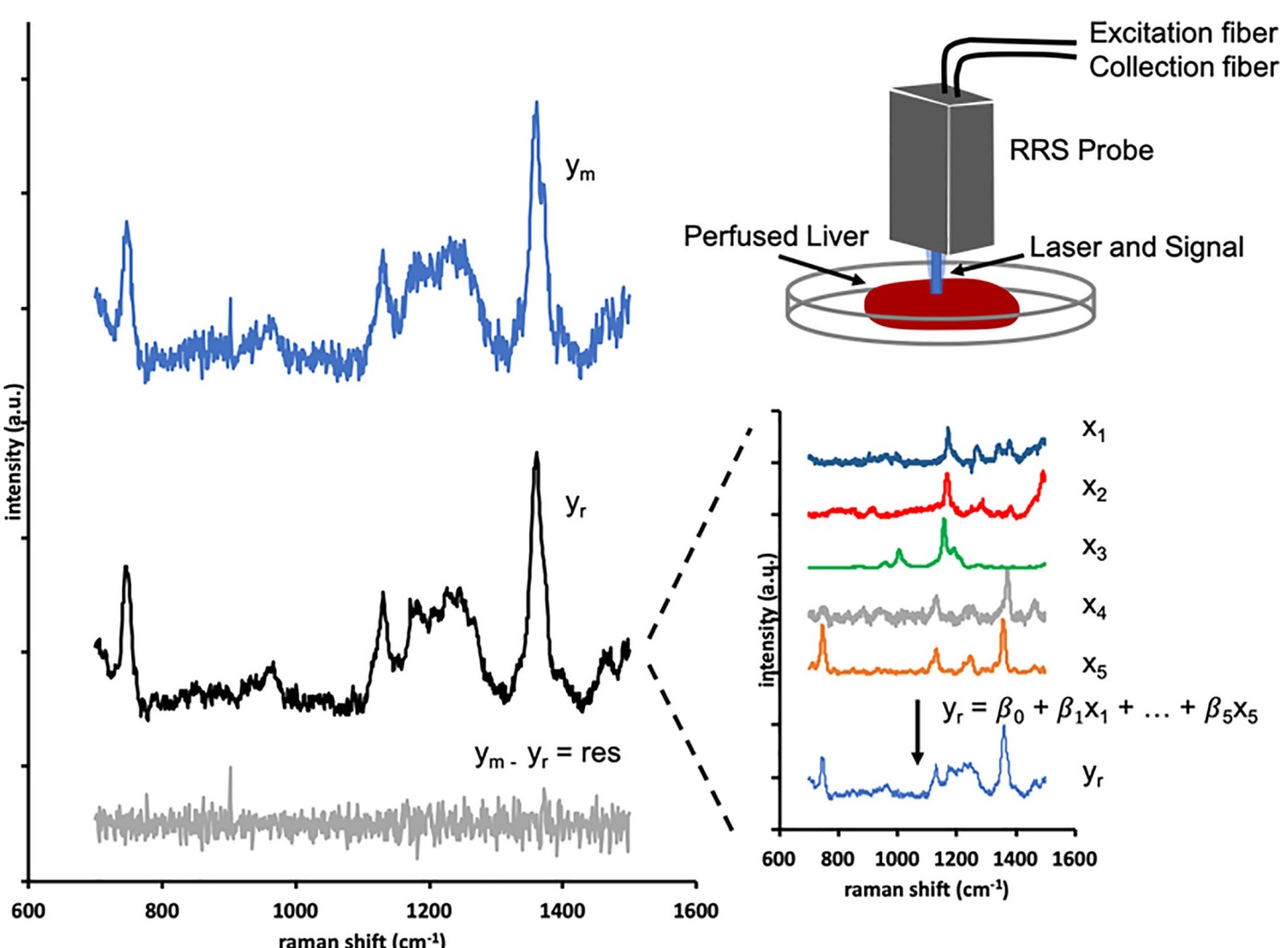

**Fig 1. Spectra library and processing algorithm.** The excitation light is focused on a 1.5 mm diameter spot on the liver tissue, and the scattered light is collected by the probe optics and coupled to the collection fiber. In the spectrometer, the light is passed through a grating onto a CCD which is read each second to produce a raw spectrum. The fluorescence background is subtracted, leaving the measured spectrum ($y_m$) for regression. The regression algorithm finds the best fit ($y_r$) as the scaled addition of the library chromophore spectra ($x_1 \dots x_n$) in order to minimize the unexplained residual spectrum ($y_m-y_r$). For perfused rat livers, the regression library included liver bile ($x_1$), perfusate ($x_2$), beta carotene ($x_3$), oxidized mitochondria ($x_4$), and reduced mitochondria ($x_5$).

Guided by the observed peaks in the complex spectra obtained from rat livers during perfusion, in combination with a literature search for molecules that are abundant in livers and have absorption peaks near the excitation wavelength of 441 nm, we identified the contributions of 5 components to the complex RR spectrum of perfused livers: oxidized mitochondria, reduced mitochondria, beta carotene, bile and the perfusate. This led to the final library as shown in Fig 1, which enabled the algorithm to accurately explain the complex whole liver spectrum with a random unexplained residual error with a root mean squared that was only 4% of the mitochondrial signal.

### The mitochondrial redox state during machine perfusion of ischemic livers

After translating the 3RMR measurement from hearts to livers, we quantified the mitochondrial redox ratio during 3 hours of subnormothermic machine perfusion (SNMP) after cold

ischemia. The clinical standard for organ preservation is hypothermic preservation (HP) at 4˚C in a specialized preservation solution [42]. For rat livers, the maximum HP duration that results in 100% transplant survival is 24 h [43]. It has been shown that extending the duration of cold ischemia (CI) during HP leads to a sharp decline in organ viability, resulting in 0% transplant survival after 72 h of HP, despite a 3-h (SNMP) resuscitation [43, 44]. Therefore, we studied the 3RMR in rat livers during SNMP after 24-h-CI (n = 6) and 72-h-CI (n = 5) as these two CI durations represent transplantable and non-transplantable rat livers, respectively [8]. Perfusion parameters of the livers are provided in S1 Fig. To account for potential spatial differences, we measured the 3RMR at three different positions and compared the average between the groups.

Directly after hypothermic preservation the mitochondria in both the 24-h-CI (n = 6) and 72-h-CI livers (n = 5) were reduced as indicated by a high 3RMR that was not different between the groups (50.68 ± 13.29 and 54.09 ± 21.59, respectively; mean ± standard deviation (SD) throughout the text; only p-values of statistically significant differences and trends are presented in text), as shown in Fig 2A and 2B. During the first half hour of perfusion the 3RMR in both the 24-h and 72-h CI livers substantially decreased (18.81 ± 7.42 and 24.12 ± 9.59, respectively at T = 30 minutes), indicating that the mitochondria became more oxidized during perfusion. However, after 30 minutes, the 3RMR of the 72-h-CI livers progressively increased over time while the 3RMR of the 24-h-CI livers decreased. This led to a significantly higher 3RMR in the 72 hours compared to the 24-h-CI livers at 60 minutes (16.75 ± 6.45 vs. 28.03 ± 7.94, p = 0.0543), 120 minutes (14.91 ± 4.28 vs 31.59 ± 2.80, p = 0.0029; Fig 2C), and 180 minutes (14.94 ± 5.97 vs 35.02 ± 13.50, p = 0.0004; Fig 2D) of perfusion (two-sided p values from repeated measures ANOVA followed by the Sidak post-hoc test for multiple testing throughout the text, unless otherwise specified). Also, 1-hr warm ischemic (WI) treatment of rat livers in saline solution produces significant injury as can be seen by high 3RMR values (S2 Fig). As expected, SNMP of the livers leads to reduction in 3RMR. This data shows that 3RMR values do not depend on the preservation modality and can be used to monitor dynamic changes in mitochondria as a function of perfusion.

## Spatial differences in the response of mitochondrial redox to ischemia reperfusion

To study potential spatial difference in the mitochondrial redox state we compared the 3RMR that we measured at the three different positions. As shown in Fig 3A, position 1 (P1) was measured on the middle of the left lateral lobe, P2 distal on the left lateral lobe, and P3 on the middle of the right medial lobe. In the 24-h-CI livers, overall the 3 locations showed the same trend with high 3RMR directly after HP that decreased during the first hour of perfusion and was followed by low and stable 3RMR during the remaining 2 hours of perfusion (Fig 3B). However, at P2 the 3RMR first increased during the first 15 minutes of SNMP and then reduced in the same way as the 3RMR at P1 and P3. As result, the 3RMR was significantly higher at P2 compared to P1 and P3 at 15 minutes (67.81 ± 5.03 vs 27.99 ± 21.69, p < 0.0001; 67.81 ± 5.03 vs 42.03 ± 4.15, p = 0.0202; respectively) and 30 minutes (44.16 ± 18.58 vs 21.80 ± 13.52, p = 0.0199; 44.16 ± 18.58 vs 20.73 ± 10.48, p = 0.0394; respectively) of perfusion.

In the 72-h-CI livers, the 3RMR at P1 and P3 followed the same trend with high 3RMR directly after HP that decreased during the first hour of perfusion but then increased during the remaining 2 hours of perfusion (Fig 3C). In contrast, the 3RMR at P2 stayed high during the first hour of perfusion and eventually reached the same values as P1 and P3 at the end of perfusion. At 60 minutes of perfusion this resulted in substantial higher 3RMR at P2

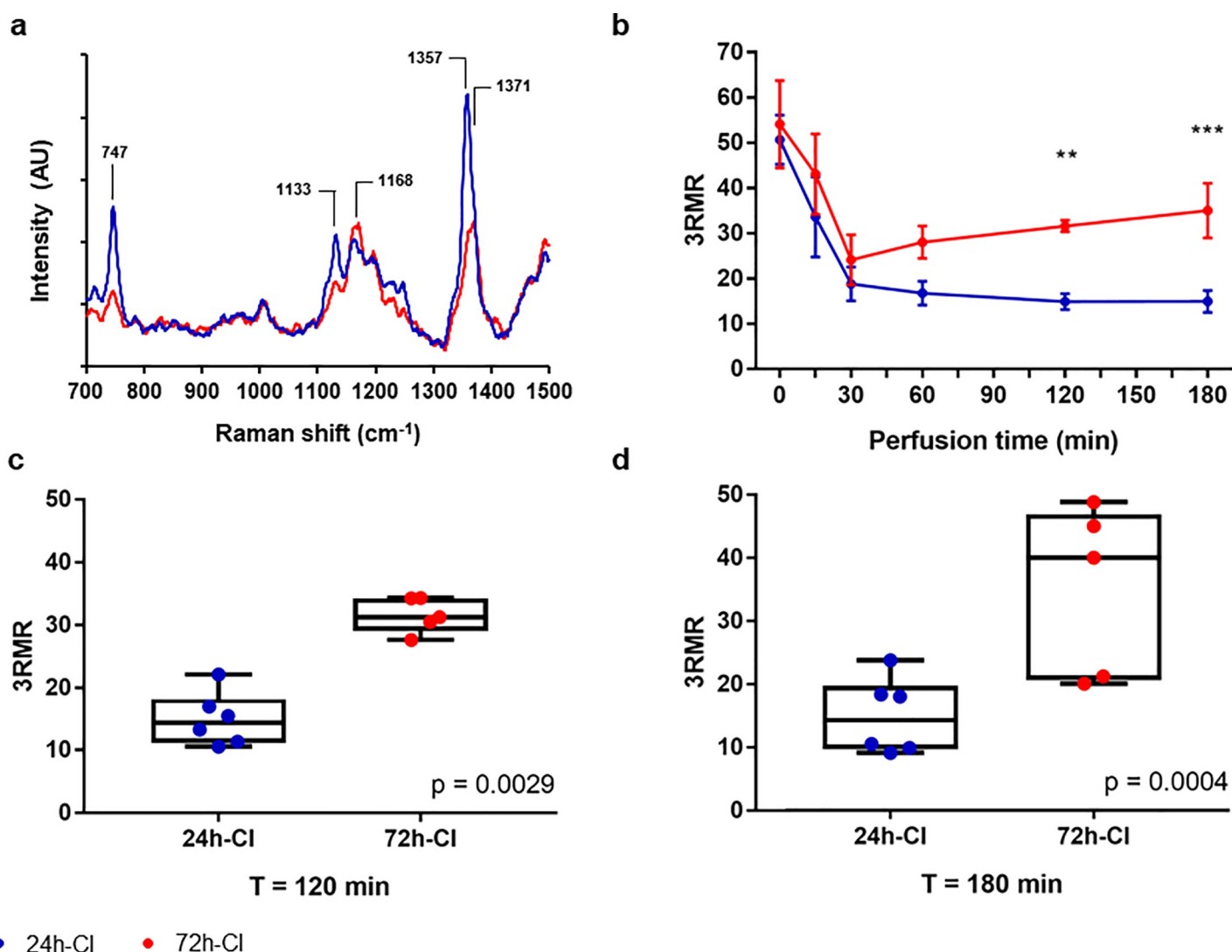

**Fig 2. The mitochondrial redox state during machine perfusion of ischemic livers.** a) Representative RR spectrum of a liver with a low 3RMR (blue) and a high 3RMR (red). Peaks for distinguishing oxidized and reduced mitochondria are identified in the figure with black lines. These peaks represent unique vibrational modes of the porphyrin ring in the mitochondrial cytochromes and result from vibrations of the individual bonds of the molecule as shown. [17]. b) 3RMR of 24 and 72 hours cold ischemic liver during 3 hours of machine perfusion, blue and red respectively. c) Data distribution of the 3RMR at 120 minutes of perfusion. d) Data distribution of the 3RMR at 180 minutes of perfusion. Stars denote statistical significance (two-way repeated measures ANOVA, followed the Sidak's post-hoc test): *$0.01 < p < 0.05$; **$0.001 < p < 0.01$; ***$0.0001 < p < 0.001$; ****$p < 0.0001$. Boxes: Median with interquartile range. Whiskers: Min max.

compared to P1 and P3 that trended toward statistical significance (48.33 ± 12.71 vs 22.14 ± 8.64, p = 0.0833; and 48.33 ± 12.71 vs 21.72 ± 10.22, p = 0.1003; respectively).

Comparing the 3RMR at each of the three positions between the 24-h-CI and 72-h-CI livers (Fig 3D–3F) showed differences between the mitochondrial redox status measured in the left lateral lobe (P1 and P2) but not in the right medial lobe (P3). At P1, the 3RMR in both experimental groups decreased during the first 30 minutes of perfusion which was followed by a continuing decrease of the 3RMR in the 24-h-CI liver while conversely the 3RMR in the 72-h-CI livers progressively increased after 30 minutes of perfusion. This resulted in a significantly higher 3RMR in the 72-h-CI than in the 24-h-CI livers at the end of perfusion (34.85 ± 22.90 vs 11.73 ± 4.38, p = 0.0122 at T = 180). At P2, the 3RMR in the 24-h-CI livers substantially decreased during the first 2 hours whereas the 3RMR at P2 stayed high during the first hour of

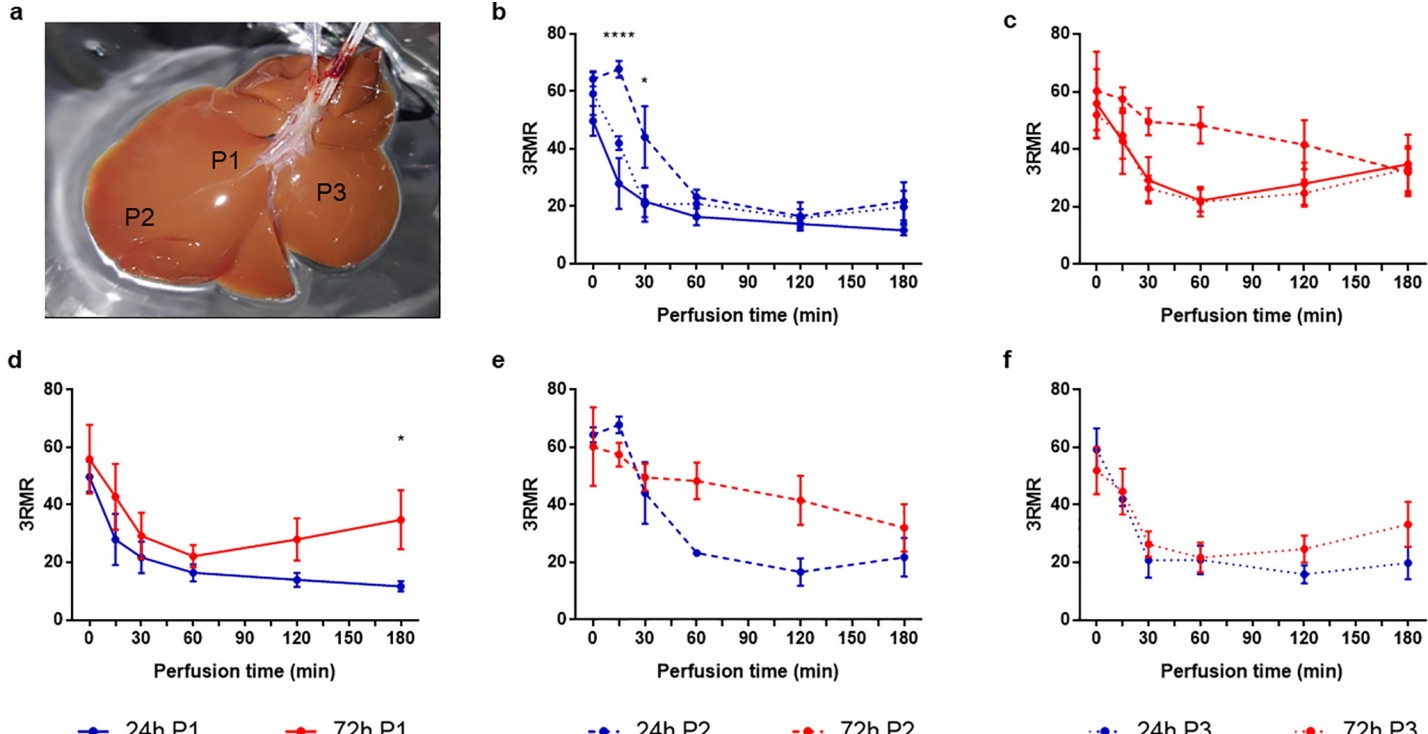

**Fig 3. Spatial differences in the response of mitochondrial redox to ischemia reperfusion.** a) Photo of a rat liver showing the position where the 3RMR was measured. Position 1 (P1, continuous lines) was measured on the middle of the left lateral lobe, P2 (dashed lines) distal on the left lateral lobe, and P3 (dotted lines) on the middle of the right medial lobe. b) Differences in the 3RMR during machine perfusion in 24-h-CI livers (blue lines). c) Differences in the 3RMR during machine perfusion in 72-h-CI livers (red lines). d) Difference between the 3RMR at P1 between 24-h-CI and 72-h-CI liver. e) Difference between the 3RMR at P2 between 24-h-CI and 72-h-CI liver. f) Difference between the 3RMR at P3 between 24-h-CI and 72-h-CI liver. Stars denote statistical significance (two-way repeated measures ANOVA, followed the Sidak's post-hoc test): $^*0.01 < p < 0.05$; $^{**}0.001 < p < 0.01$; $^{***}0.0001 < p < 0.001$; $^{****}p < 0.0001$. Dots: Means. Error bars: SEM.

perfusion. This led to substantial differences in 3RMR between the 24-h-CI and 72-h-CI livers at 60 and 120 minutes of perfusion that trended towards significance ($23.25 \pm 1.43$ vs $48.33 \pm 12.71$ p = 0.0709; and $16.63 \pm 8.23$ vs $28.02 \pm 16.39$ p = 0.0727; respectively) At P3, the 3RMR in both the 24-h-CI and 72-h-CI livers followed the same trajectory with an decrease in 3RMR during the first 30 minutes of perfusion and low 3RMR during the remaining perfusion. No significant differences nor trends towards significance were observed between the 3RMR at P3 in the 24-h-CI and 72-h-CI livers at any of the timepoints during perfusion.

## Correlations between the mitochondrial redox state and oxygen uptake perfusion

Complex IV in the ETC accounts for >95% of oxygen consumption and its function is directly related to the redox status of cytochrome a,a3 [17, 22]. The oxygen uptake rate of the 24-h-CI and 72-h-CI livers during machine perfusion is shown in Fig 4A and 4B. We hypothesized that the oxygen uptake rate during machine perfusion would be inversely correlated to the 3RMR which would indicate that mitochondria with a more reduced redox status consume less oxygen. Indeed, the oxygen uptake rate of the livers was negatively correlated to the corresponding 3RMR at the discrete timepoints (Fig 4A). The correlation was moderately strong (r = -0.6566) and highly significant (p < 0.0001). While the means of the oxygen uptake rate were higher in the 24-h-CI than in the 72-h-CI livers at every time point, no statistically significant differences were observed between the two groups. Thus, to further augment this analysis with a more direct

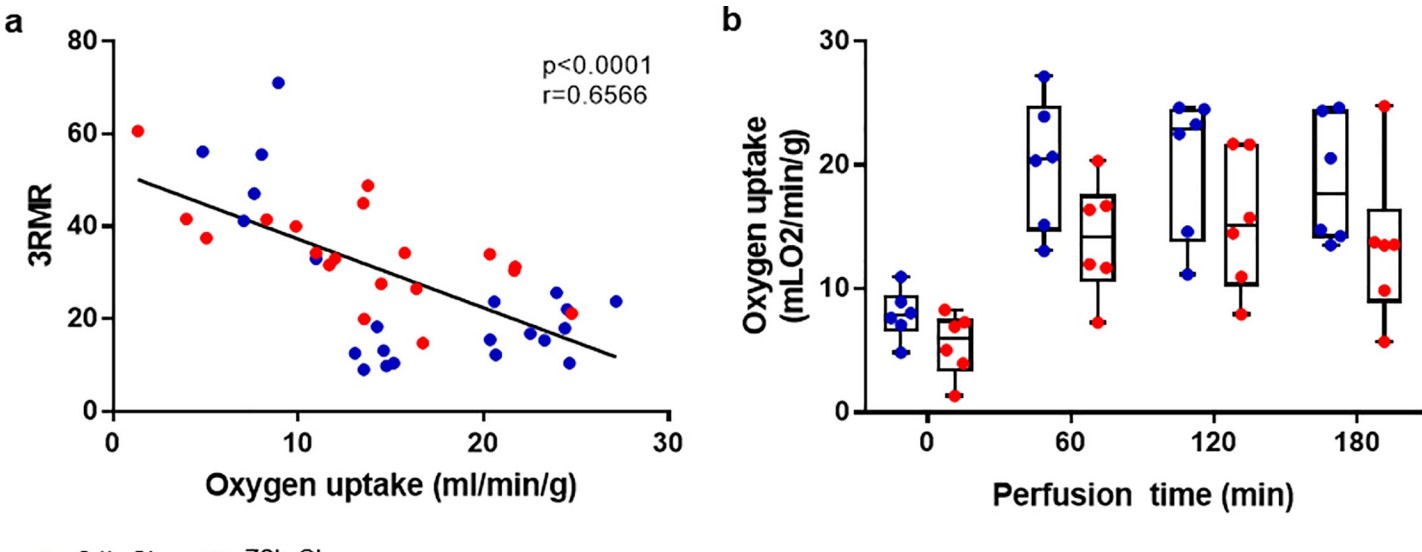

**Fig 4. Mitochondrial redox state and oxygen uptake.** a) Scatterplot of the 3RMR versus the oxygen uptake rate of 24-h-CI (blue) and 72-h-CI (red) livers. Black lines: Fit of linear regression with 1, 39 degrees of freedom (DFn, DFd). r: Pearson's correlation coefficient. b) Boxplots of the oxygen uptake rate of the livers during machine perfusion. Boxes: Median with interquartile range. Whiskers: Min max.

measure of mitochondrial function, a JC-1 mitochondrial membrane potential (MMP) assay was performed with primary rat-hepatocytes. Plates with primary rat-hepatocytes were stored for 24- or 72-hours at 4˚C followed by 3-hours of recovery as described in the methods section. Corresponding to the 3RMR values at the end of perfusion, the red/green fluorescence ratio of JC-1 dye which is directly proportional to the MMP, declines significantly for the 72-hour cold ischemic livers compared to the 24-hour cold ischemic livers (p < 0.0001) as shown in S3 Fig.

### Differences in the cellular redox state

The cellular redox status changes as a result of IRI which can be assessed by the NAD+/NADH and NADP+/NADPH ratios. These ratios have been previously evaluated in attempt to assess IRI during liver transplantation. Although IRI pathways of the cellular and mitochondrial redox states are interconnected, the cellular redox state does not necessarily correspond to the mitochondrial redox status. To evaluate the relation between the cellular and mitochondrial redox status, we additionally measured the NAD+/NADH and NADP+/NADPH redox ratios with mass spectrometry in flash frozen tissue samples taken at the end of perfusion (Fig 5A and 5B, respectively). Conversely to the mitochondrial redox state, neither the NAD/NADH ($0.015 \pm 0.005$ vs $0.023 \pm 0.009$, respectively) nor the NADP+/NADPH ($0.436 \pm 161$ vs $0.462 \pm 0.087$, respectively) were significantly different between the 24-h-CI and 72-h-CI livers. A similar trend was observed for the lipid peroxidation and protein oxidation levels (S4 Fig) as seen by the concentration of malondialdehyde (MDA) ($2.446 \pm 0.369$ vs $2.163 \pm 0.291$ nmoles/mg protein, respectively) and protein carbonyl content ($3.839 \pm 1.19$ vs $3.227 \pm 0.907$ nmoles/mg protein, respectively). We hypothesize this is because lipid and protein oxidation levels can be influenced by several factors whereas 3RMR is a more direct measure of mitochondrial redox state.

### Discussion

IRI is major consideration for any liver surgery whereby the blood supply to the liver is obstructed [1, 2]. Especially in liver transplantation, IRI is a critical problem that can lead to

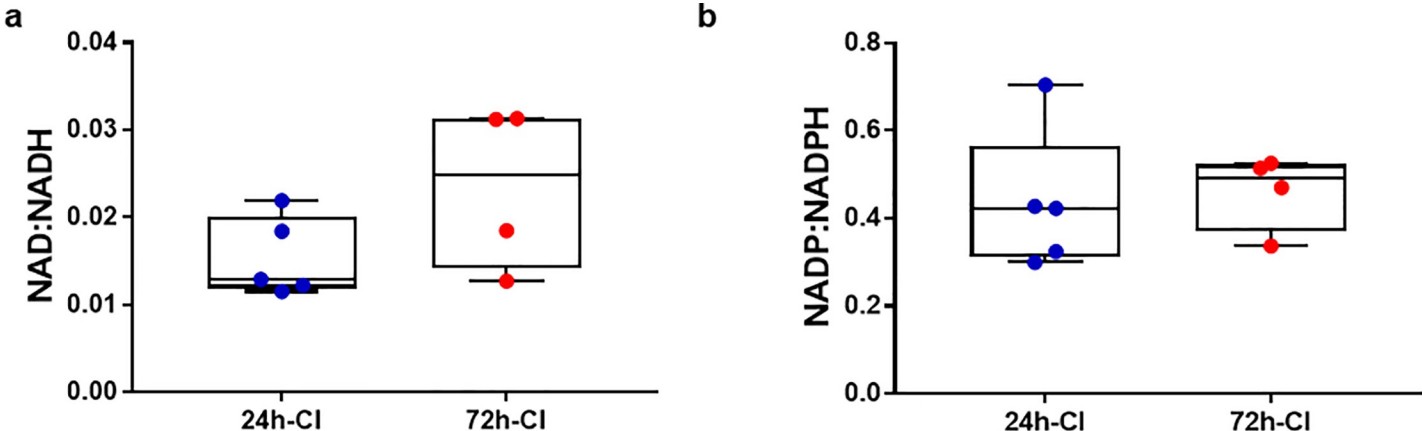

**Fig 5. Cellular redox state after machine perfusion.** a) Nicotinamide adenine dinucleotide (NAD) redox ratios in tissue of 24-h-CI (blue) and 72-h-CI (red) livers. b) Nicotinamide adenine dinucleotide phosphate (NADP) redox ratios. Boxes: Median with interquartile range. Whiskers: Min max.

life threatening complications and substantially limits the availability of organs for transplantation [3, 4]. However, there are no early diagnostics of hepatic IRI and the fulminant injury often only becomes evident upon reperfusion. Mitochondria play a pivotal role in all pathways that lead to IRI and thus are an ideal diagnostic target [2, 14, 15]. During ischemia, lack of oxygen curtails the redox reactions in the mitochondrial membrane complexes that subsequently become increasingly reduced and forms the first link in the chain of events that lead to IRI [11, 17]. In this study we translated the resonance Raman reduced mitochondrial ratio (3RMR), originally developed to assess cardiac ischemia, to provide a rapid, non-invasive, and label-free diagnostic that quantifies the mitochondrial redox status in whole livers during machine perfusion.

The 3RMR is quantified based on the relative strength of the spectrum of oxidized versus the spectrum of reduced mitochondria in the complex spectrum of the liver. The redox status of cytochrome a,a3 in mitochondrial complex IV was of particular importance as it forms the active site for oxygen in the electron transport chain. Cytochrome a,a3 is known to exhibit strong resonance enhancement at the used excitation wavelength of 441 nm. Hence, it substantially contributes to the library spectra of reduced and oxidized mitochondria that we used to calculate the 3RMR. Cytochrome b-c1 and cytochrome c are also part of the ETC and both have absorption peaks within proximity of the used excitation wavelength [45]. Therefore, they likely exhibit some resonance enhancement and partially contribute to the 3RMR.

Accuracy of the 3RMR relies on the assumption that the library spectra are the only important contributors to the complex RR spectrum that is obtained from the livers. Therefore, it is essential to identify other molecules in the liver that exhibit substantial resonance enhancement at the 441nm excitation wavelength. We identified beta Carotene, bile and the perfusate to be important contributors to the complex RR liver spectra. Beta-carotene is likely to resonate at 441nm because it has an absorption peak at 448 nm [46]. Also known as provitamin A, it has high concentration is the liver where it is used for vitamin A synthesis [47]. The RR spectrum of bile is likely caused by bilirubin which has an absorption maximum at 440 nm [48]. Bilirubin is the metabolite of hemoglobin that is broken down by the liver and excreted in the bile and therefore is expected to have higher concentrations in the liver. Further, the perfusate contains a plethora of salts, saccharides, proteins, amino acids and vitamins that were captured in a composite RR spectrum of the perfusate and used as library component. Taken together, we are confident that the library spectra that we identified are the only substantial contributors

to the complex liver spectrum because of the random nature and low amplitude of the residual error. This is witnessed by the low root mean square of the residual error that approaches the shot noise which is the theoretical minimum error.

We found that the 3RMR was high in rat livers after exposure to different durations of cold ischemia and decreased as soon livers were perfused. This indicates that the mitochondria are relatively reduced during cold ischemia and become oxidized during subsequent re-oxygenation during perfusion. This agrees with existing literature and gives us further confidence in the validity of the 3RMR to quantify the mitochondrial redox state. Interestingly, we found that the mitochondria in mildly and severely ischemic livers were equally reduced after ischemia and that the initial response to oxygen exposure during perfusion was similar during the first 30 minutes of perfusion. Whereas the mitochondria of mildly ischemic livers remained oxidized throughout perfusion, the 3RMR of severely ischemic livers progressively increased after perfusion. This delineated the mild from severe ischemic livers with high statistical significance. A similar mitochondrial response to ischemia-reperfusion was previously observed in transplanted human livers. Livers that developed early allograft dysfunction initially demonstrated the same increase in adenylate energy charge (EC) as successfully transplanted livers during reperfusion but subsequently failed to maintain a high EC [49]. Although the EC is considered one of the most comprehensive liver viability parameters, it cannot be clinically used because determination of the EC in the biopsied tissue takes several days [49–52]. In this regard, it is promising that we observed a similar hepatic mitochondrial response using a real-time noninvasive method.

Alternative to the EC, the redox ratios of cofactors NAD and NADP have previously been studied as a marker of mitochondrial function. However, no significant correlations with liver viability were observed [49]. Although NAD and NADP have spectroscopic properties that can be leveraged for rapid measurement of the NAD/NADH and NADP/NADPH redox ratios, these mostly reflect the cellular redox state and are not specific to mitochondria [31]. This agrees with the findings in the present study where we found significant differences in the mitochondrial redox state between transplant and non-transplantable livers, but not in the cellular redox state as measured by NAD/NADH and NADP/NADPH ratios as well as lipid and protein oxidation levels. This suggests an important advantage of the 3RMR that is specific to the mitochondrial redox status. The mitochondrial redox status–in particular that of cytochrome a,a3 –is directly related to oxygen uptake by mitochondria that is >95% of the cellular oxygen consumption [17, 22]. In accordance, we found a significant negative correlation between the oxygen uptake and the 3RMR (Fig 4A). However, it should be noted that statistically significant differences for oxygen uptake were not achieved when comparing 24- and 72-hour groups at a given time point during perfusion (Fig 4B), suggesting complex factors dictate oxygen consumption rates during perfusion. As a result, we confirmed differences in 24- and 72-hour ischemic livers through a more direct measurement of the mitochondrial membrane potential in an *in vitro* assay using primary rat hepatocytes.

While the averaged 3RMR were different between mild and severely ischemic livers, we found spatial differences in the mitochondrial redox status after ischemia. We hypothesize these observations may be caused by differences in perfusion. First, in both the mild and severe ischemic livers the redox status of the mitochondria in peripheral regions lagged to the mitochondria in more proximal anatomical locations. In general, peripheral regions are less well perfused that may explain this observation. Second, the mitochondrial redox response after ischemia was significantly different between the mild and severe ischemic livers in the left lateral lobe but the same in the right medial lobe. Indeed literature reports better perfusion of the right medial lobe in anatomical and physiological studies [53–55]. From a practical perspective, this observation demonstrates the significance to account for spatial differences in

viability assessment of the liver. This not only applies to the quantification of the 3RMR in human livers but likely also to other spatial dependent viability metrics such as histological, metabolomic or proteomic analyses [8].

One limitation is that we evaluated the mitochondrial redox state during oxygenated subnormothermic machine perfusion (21˚C). Although SNMP has the advantage that the organ is metabolically active while an oxygen carrier can be omitted [56, 57], it remains in pre-clinical phase as current clinical trials are conducted with either hypothermic (10˚C) or normothermic (37˚C) perfusion temperatures [37–39]. We are confident that the diagnostic developed in the present study can be directly applied to human livers undergoing HMP. However, for application to livers that are perfused with blood such as during normothermic machine perfusion or *in vivo*, we anticipate that the spectra of oxidized and reduced hemoglobin should be included in the library. Hemoglobin is known to exhibit strong resonance enhancement at the excitation wavelength of 441 nm and will considerably contribute to the complex RR spectrum of the liver. Feasibility of measuring the 3RMR in blood perfused hearts *in vivo* was recently demonstrated [17].

This work provides a novel quantitative diagnostic of the hepatic mitochondrial redox status via real-time, non-contact and label-free readings on the liver surface. The results demonstrate that this diagnostic modality can be used to significantly distinguish livers with mild from livers with severe ischemic injury during oxygenated machine perfusion and discover spatial differences in the mitochondrial redox response to hepatic ischemia reperfusion. Ultimately, the 3RMR may be used in future to predict the viability of human livers during transplantation and other major liver surgery and to provide better understanding of hepatic IRI. While further adjustments to the library may be necessary for the translation of this technology in the humans, the current work should provide a strong reference for conducting such experiments.

## Materials and methods

### Liver procurement

The animals were maintained in accordance with National Research Council guidelines and the experimental protocols were approved by the Institutional Animal Care and Use Committee (IACUC) of Massachusetts General Hospital (Prot#2017N000227). The livers were procured under complete anesthesia by isoflurane. The liver was exposed through a transverse abdominal incision and freed from its connecting ligaments. The bile duct and portal vein were cannulated with 24-gauge and 16-gauge catheters, respectively and the hepatic artery was ligated. Directly after cannulation the infrahepatic and suprahepatic inferior vena cava were transected and the livers were directly flushed with 60 ml of ice-cold saline. During the flush, the rats were euthanized as results of the exsanguination. After the livers were removed from the abdomen, the livers were flushed with 30ml ice-cold University of Wisconsin (UW) solution (Bridge to Life, Columbia, SC, USA). Next, the livers were submerged in 50 ml fresh UW solution and stored at 4˚C in a sealed bag.

### Subnormothermic machine perfusion

We recently described the details about the machine perfusion protocol elsewhere [8]. In summary, the machine perfusion system provides an oxygenated, temperature (21˚C ± 1˚C), pressure and flow controlled, non-pulsatile circulation via the portal vein. During the first 30 min of perfusion, the perfusion pressure was gradually increased to 5 mmHg that was maintained throughout perfusion unless the maximum 25 ml/min was reached.

The perfusate consisted of Williams Medium E solution (Sigma-Aldrich, St Louis, MO, USA) prepared according to the manufacturer's instructions. The perfusate was supplemented with dexamethasone (24 mg/l; Sigma-Aldrich), insulin (5 U/l; MGH Pharmacy), heparin (2000 U/l; MGH Pharmacy), and bovine serum albumin (10 mg/ml; Sigma-Aldrich). Prior to perfusion of the liver, the perfusate was circulated for ~15 min in the perfusion system to oxygenate and if necessary the pH was adjusted to 7.3–7.4 with sodium bicarbonate.

## RRS device description

The RRS system (Pendar Technologies, Cambridge, MA) consists of a laser pump source, source/collection fiber optic probe, and high resolution/high throughput spectrometer. The 441 nm laser pump source provides 4 mW of single line excitation light to the source fiber. A custom-built probe focuses a 1.5 mm diameter laser spot 9 mm from the probe tip and collects the scattered light from the sample area. Light is passed through a filter to eliminate the excitation signal, and the Raman signal is returned through a second fiber to the spectrometer. The spectrometer has a full width at half maximum resolution of 8 cm$^{-1}$, and an internal Stokes shift calibration reference (acetaminophen) which achieves an absolute Stokes shift accuracy of <0.4 cm$^{-1}$. Cosmic rays are eliminated from CCD pixels, and the temperature-controlled image sensor maintains a constant dark current. Resulting spectra are captured approximately every 1 second and signal averaged over a rolling 180 second window to create the raw spectrum for analysis. A probe holder was designed to help position the probe over the liver during measurements.

The readout from the CCD is recorded and analyzed using software developed in LabView. After the spectrum is adjusted for cosmic rays and dark current, an estimate of the fluorescence baseline is made and subtracted from the signal, leaving the Raman spectrum. A regression analysis is performed against a stored library of known spectra in order to find the best fit. The resulting fit coefficients represent the relative concentrations of chromophores in the tissue, and the coefficients for oxidized and reduced states can be ratioed in order to produce an absolute result for saturations or redox states. Heme containing structures have a strong absorption band between 400–450 nm (the Soret Band), and excitation near this band results in resonant enhancement of the Raman signal. A 441 nm excitation wavelength was selected to optimize the signal from mitochondrial cytochromes, as there was minimal residual hemoglobin in the non-blood perfused livers.

The readout from the CCD produces a known noise level for each recorded spectrum (shot noise). The residual produces a lower limit of explainable spectrum in the signal. For the rat liver measurements, the shot noise was approximately 3.6% of the mitochondrial signal level. The root mean square (RMS) error of the residual was 4% of the mitochondrial signal, which approached the theoretical limit.

## Creation of RRS library

Based on experiments with in-vivo myocardial tissue, Pendar had previously developed a library of chromophores in the oxidized and reduced state [17], and the mitochondrial library was applied to the liver measurements. To obtain the library spectra, fresh mitochondria were isolated from homogenized rodent hearts and purified by a process of serial differential centrifugation. Oxidized conditions were established by exposing mitochondria to hyperoxia (100% oxygen flow) in an abundance of substrate (succinate and ADP) and verified based on the absence of a reduced RR marker peak at 1357 cm$^{-1}$. Fully reduced conditions were established by exposure to 100% nitrogen with dissolved sodium dithionate within a sealed cuvette and verified by absence of the oxidized marker peak at 1371 cm$^{-1}$.

Pilot experiments with rat livers identified additional peaks not explained by the oxidized or reduced mitochondria spectra. We determined that bile, beta carotene, and the perfusate contributed additional peaks to the spectra and thus were needed in the regression library in order to fully explain the measured signal. To create new library spectra, a sample of the material was placed in a 1 mL glass vial, sealed, and positioned in front of the RRS probe inside of a dark box to eliminate light contamination. A spectrum was collected for 600 seconds and stored. The fluorescence background was subtracted, and the resulting spectrum stored as an additional library for use in the regression. Bile was obtained directly from rat livers during machine perfusion. Beta carotene (Sigma Aldrich) was dissolved in oil prior to creating the library. Perfusate samples were taken from the perfusate reservoir at the time of performing an experiment.

## Experimental design of the ex vivo perfusion experiment

We quantified the mitochondrial redox ratio during 3 hours of subnormothermic machine perfusion (SNMP) after two durations of cold ischemia (CI). The clinical standard for organ preservation is hypothermic preservation (HP) at 4˚C in a specialized preservation solution [42]. For rat livers, the maximum HP duration that results in 100% transplant survival is 24 h [43]. It has been shown that extending the duration of CI during HP leads to a sharp decline in organ viability, resulting in 0% transplant survival after 72 h of HP, despite a 3-h (SNMP) resuscitation [43, 44]. Therefore, we studied the 3RMR in rat livers during SNMP after 24-h-CI (n = 6) and 72-h-CI (n = 5) as these two CI durations represent transplantable and non-transplantable rat livers, respectively [8]. We measured the 3RMR just before perfusion (T = 0) and at 15, 30, 60, 120, and 180 minutes of perfusion. At each timepoint we measured the 3RMR at three different positions. These were on the middle of the left lateral lobe (P1), distal on the left lateral lobe (P2), and on the middle of the right medial lobe (P3), as shown in Fig 3A. For each time point, we compared the means of the 3 locations to account for spatial differences. We also compared the 3 locations both within and between each experimental group to evaluate the spatial differences in the mitochondrial redox state. For inducing warm ischemic injury, rat livers were stored in a saline solution and incubated at 37˚ C in a water bath for 1 hour (n = 3). This was followed by perfusion and 3RMR measurement before perfusion (T = 0) and at 30, 60, 90, and 120 minutes of perfusion and 3RMR measurement.

## Perfusion data acquisition and processing

The liver was weighed directly after procurement and machine perfusion. Perfusion pressures and flowrates in the portal vein were recorded every 60 minutes of perfusion. The partial oxygen pressure was measured in the portal vein (inflow) and in the intrahepatic vena cava (outflow) and the potassium concentration was measured only in the outflow using an i-STAT blood analyzer (Abbott Laboratories, Chicago, IL, USA). To calculate the portal vein vascular resistance, the flowrate was first divided by the weight of the liver after procurement and subsequently the perfusion pressure was divided by the corrected flowrate. The oxygen uptake was calculated using the partial oxygen pressures and flowrates of the inflow and outflow as we recently described in detail elsewhere [8]. The oxygen uptake was divided by the weight of the liver after procurement.

## Mitochondrial membrane potential measurement

Freshly isolated primary rat hepatocytes were seeded in C+H medium at 131,500 cells per cm^2 in collagen coated 24 well plates and allowed to attach for 24 hours. For fresh control, cells were washed with PBS and treated with 7 μM JC-1, 25 μM verapamil, and 2 drops/mL

NucBlue® Live ReadyProbes® Reagent in C+H medium for 30 minutes at 37C, 10% CO2. For positive control, cells were washed with PBS and treated with 7 μM JC-1, 25 μM verapamil, 20 μM CCCP, and 2 drops/mL NucBlue® Live ReadyProbes® Reagent in C+H for 30 minutes. After incubation, cells were washed with PBS and imaged on a Zeiss Axiovert 200 M Inverted microscope using DAPI, FITC, and RFP filters. For static cold storage experimental groups, cells were washed in PBS followed by the addition of University of Wisconsin Solution for 24 or 72 hours at 4C. Following static cold storage, cells were washed with PBS and replaced with C+H medium for 3 hours and imaged with JC-1 as described above. To quantify mitochondrial membrane potential, the ratio of red to green fluorescence was quantified using ImageJ for fresh cells, CCCP control, and cells exposed to UW for 24 and 72 hours. For viability assessment, the same groups were washed with PBS and exposed to 2 drops/mL NucBlue® Live ReadyProbes® Reagent, 4 μM Calcein-AM, and 2 μM ethidium homodimer-1 in serum-free medium for 25–30 minutes. Cells were then washed with PBS, replaced with C+H medium, and immediately imaged using DAPI, FITC, and RFP filters.

### Co factor analysis

For n = 5 24-h-CI and n = 4 72-h-CI livers we measured the NAD, NADH, NADP, NADPH concentrations in liver tissue that was flash frozen in liquid nitrogen immediately after perfusion. This analysis was performed by the mass spectrometry core of Shriners Hostpitals for Children (Boston, MA, USA) as described in detail elsewhere [56].

### Lipid peroxidation and protein oxidation analysis

For n = 4 24-h-CI and n = 3 72-h-CI livers we measured lipid peroxidation marker malondialdehyde (MDA) using Lipid Peroxidation (MDA) Assay Kit (MAK085) by Millipore Sigma and the protein oxidation marker protein carbonyl with a Protein Carbonyl Content Assay Kit (ab126287) by Abcam according to the manufacturer's instructions. Total protein content was measured with the Pierce™ Rapid Gold BCA Protein Assay Kit by ThermoFisher Scientific.

### Statistical analysis

We performed all statistical analyses in Prism 7.03 (GraphPad Software Inc., La Jolla, CA). Repeated measures one-way ANOVAs were used for the comparison of the time-course mean 3RMR and perfusion data. Repeated measures two-way ANOVAs were used to compare the 3RMR data measured at the 3 different positions. In both cases the Sidak post hoc test was used to test for statistical differences between the experimental groups at discrete time points and to correct for multiple comparisons. Linear regressions in combination with Pearson's correlation coefficient were used to correlate the oxygen uptake rate during to the 3RMR during machine perfusion and the correlations. The NAD/NADH, NADP/NADPH ratios, MDA, and protein carbonyl levels were compared with unpaired Welch's t tests. In all cases, two-sided p-values were considered a trend when <0.10 and statistically significant when <0.05.

### Supporting information

**S1 Fig. Perfusion parameters of cold ischemic livers.** a) Vascular resistance in the portal vein. b) Flowrate of the portal vein. c) pressure in the portal vein. d) Weight gain at the end of perfusion compared to the liver weight after procurement. e) Oxygen uptake rate (OUR) f) potassium concentrations in the intrahepatic inferior vena cava. Dots: means. Error bars: SEM. Boxes: Median with interquartile range. Whiskers: min & max. Cones: individual data

points.
(DOCX)

**S2 Fig. 3RMR measurements and perfusion parameters of warm ischemic livers.** To ascertain the effect on 3RMR measurements via a method of injury other than cold storage, we performed three experiments where rat livers were subjected to 1 hour of warm ischemia in a saline solution followed by 2 hours SNMP. a) A similar trend in 3RMR measurements upon injury followed by recovery as seen for cold ischemic livers. b) The high values for 3RMR at t = 0 min. c) Low values of 3RMR at t = 120 min. d) Oxygen uptake rate (OUR) over the duration of perfusion. e) 3RMR versus OUR. The line shows the trend. f) Potassium concentration. Dots: means. Error bars: SEM. Bar graph: Mean. Cones: individual data points.
(DOCX)

**S3 Fig. Mitochondrial membrane potential in sandwich cultures using JC-1 dye.** Primary rat hepatocytes cultured in a sandwich format with collagen I gel show a marginal drop in the mitochondrial membrane potential (MMP) after 24 hours of cold ischemic storage and 3 hours of recovery when compared to fresh hepatocytes, as opposed to a highly significant drop in MMP after 72 hours of cold ischemic storage and 3 hours of recovery. Hepatocytes treated with Carbonyl cyanide 3-chlorophenylhydrazone (CCCP) which disrupts the mitochondrial membrane are used as a positive control. Each bar is for n = 3. (* p = 0.0278; **** p<0.0001).
(DOCX)

**S4 Fig. Lipid peroxidation and protein oxidation assays.** a) Malondialdehyde (MDA) concentration in tissue biopsies from 24-Hr and 72-Hr cold storage and 3-hour perfusion rat livers. b) Protein carbonyl concentration in tissue biopsies from 24-Hr and 72-Hr cold storage and 3-hour perfusion rat livers. p values of 0.3098 and 0.4751 for MDA level and protein carbonyl content between the two treatment conditions indicate a statistically non-significant difference. Boxes: median with interquartile range. Whiskers: min & max.
(DOCX)

## Acknowledgments

We thank the Shriners Hospitals for Children–Boston Mass Spectrometry Core Facility and Morphology and Imaging Facility for their services.

## Author Contributions

**Conceptualization:** Reinier J. de Vries, Padraic Romfh, Shannon N. Tessier.

**Data curation:** Reinier J. de Vries, Stephanie E. J. Cronin, Padraic Romfh, Peili Chen, Shannon N. Tessier.

**Formal analysis:** Reinier J. de Vries, Stephanie E. J. Cronin, Padraic Romfh, Casie A. Pendexter, Rohil Jain, Benjamin T. Wilks, Siavash Raigani, Thomas M. van Gulik, Korkut Uygun, Shannon N. Tessier.

**Funding acquisition:** Padraic Romfh, Korkut Uygun, Shannon N. Tessier.

**Investigation:** Reinier J. de Vries, Stephanie E. J. Cronin, Padraic Romfh, Casie A. Pendexter, Rohil Jain, Benjamin T. Wilks, Siavash Raigani, Thomas M. van Gulik, Peili Chen, Heidi Yeh, Korkut Uygun, Shannon N. Tessier.

**Methodology:** Reinier J. de Vries, Peili Chen, Heidi Yeh, Shannon N. Tessier.

**Project administration:** Shannon N. Tessier.

**Resources:** Shannon N. Tessier.

**Software:** Peili Chen.

**Supervision:** Thomas M. van Gulik, Heidi Yeh, Korkut Uygun, Shannon N. Tessier.

**Validation:** Reinier J. de Vries, Shannon N. Tessier.

**Visualization:** Reinier J. de Vries, Padraic Romfh, Peili Chen, Shannon N. Tessier.

**Writing – original draft:** Reinier J. de Vries, Padraic Romfh, Shannon N. Tessier.

**Writing – review & editing:** Reinier J. de Vries, Stephanie E. J. Cronin, Padraic Romfh, Casie A. Pendexter, Siavash Raigani, Thomas M. van Gulik, Peili Chen, Heidi Yeh, Korkut Uygun, Shannon N. Tessier.

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
