## [Decision Letter · Decision Letter 0]

26 Mar 2021

PONE-D-21-06813

Non-invasive Quantification of the Mitochondrial Redox State in Livers During Machine Perfusion.

PLOS ONE

Dear Dr. Tessier,

Thank you for submitting your manuscript to PLOS ONE. After careful consideration, we feel that it has merit but does not fully meet PLOS ONE’s publication criteria as it currently stands. Therefore, we invite you to submit a revised version of the manuscript that addresses the points raised during the review process.

Your manuscript was reviewed by two experts and we received positive feedback. However, both of them asked few sound questions , which require your attention during revision.

We look forward to receiving your revised manuscript.

Kind regards,

Partha Mukhopadhyay, Ph.D.

Academic Editor

PLOS ONE

Journal Requirements:

'We have read the journal's policy and the authors of this manuscript have the following

competing interests: Drs. de Vries, Uygun, Tessier, and Chen, and Mr. Romfh have

provisional patent applications relevant to this study. Dr. Uygun has a financial interest

in Organ Solutions, a company focused on developing organ preservation technology.

P.R. and P.C. are employees and shareholders of Pendar Technologies. All competing

interests are managed by the MGH and Partners HealthCare in accordance with their

conflict of interest policies, except for P.R. and P.C. who are subject to the Research

Integrity Policy of Pendar Technologies. All other authors do not have competing

interest.'

We note that one or more of the authors are employed by a commercial company: Pendar Technologies.

3. We note that you have a patent relating to material pertinent to this article. Please provide an amended statement of Competing Interests to declare this patent (with details including name and number), along with any other relevant declarations relating to employment, consultancy, patents, products in development or modified products etc. Please confirm that this does not alter your adherence to all PLOS ONE policies on sharing data and materials, as detailed online in our guide for authors http://journals.plos.org/plosone/s/competing-interests by including the following statement: "This does not alter our adherence to  PLOS ONE policies on sharing data and materials.” If there are restrictions on sharing of data and/or materials, please state these. Please note that we cannot proceed with consideration of your article until this information has been declared.

Additional Editor Comments (if provided):

Reviewers' comments:

Reviewer's Responses to Questions

**Comments to the Author**

1. Is the manuscript technically sound, and do the data support the conclusions?

Reviewer #1: Partly

Reviewer #2: Yes

2. Has the statistical analysis been performed appropriately and rigorously? 

Reviewer #1: Yes

Reviewer #2: Yes

3. Have the authors made all data underlying the findings in their manuscript fully available?

Reviewer #1: Yes

Reviewer #2: Yes

4. Is the manuscript presented in an intelligible fashion and written in standard English?

Reviewer #1: Yes

Reviewer #2: Yes

5. Review Comments to the Author

Reviewer #1: The present study “Non-invasive Quantification of the Mitochondrial Redox State in Livers during Machine Perfusion” is a pioneer discussion about a new method for evaluating liver graft viability. It has good practical merit. Here are some major concerns to address:

1. The study compared short and long preserved livers as viable and unviable grafts. However, preservation itself may change the mitochondrial integrity. The results in this study is thus only indicative of the length of preservation. To better define the methods provided by this study, the authors need to compare a well preserved graft with an inviable liver graft destroyed in at least another way.

2. The authors removed the interference from carotene and bile. How do the authors decide the amplitude of these parameters (the beta constants in Figure 1)? However, lab rats have very consistent living condition to make these two parameters almost the same between different individuals. Plus there is no gallbladder, which contributes to the consistency of bile content in the liver. Human do not live in such conditions and everyone’s diet will change these settings. How do the authors provide an accurate array of parameters to apply to each liver graft?

3. The authors explained the present method as a measurement of mitochondrial redox condition. However, Figure 4B did not show a significance between viable and unviable grafts in mitochondrial function. This is a major inconsistency. The authors need to explain it. Also the authors need directly measure the mitochondrial function and membrane potential whether or not it is invasive.

4. The authors should analyze lipid or protein peroxidation levels in Figure 5.

Reviewer #2: In the present manuscript de Vries et al. investigated the capability of resonance Raman spectroscopy to measure mitochondrial redox status in livers during oxygenated machine perfusion to assess liver damage in a rapid, non-invasive and label-free way. IRI is a major consideration for liver surgery and remains a critical problem for transplantation contributing to limited availability of transplantable livers, therefore finding new diagnostics to identify transplantable organs remain crucial. In this study they adapted and translated the heart specific 3RMR (resonance Raman reduced mitochondrial ratio) and showed that this diagnostic modality can distinguish transplantable versus non-transplantable ischemic rat livers during subnormothermic machine perfusion after cold ischemia.

The authors should be commended for adapting the 3RMR to accurately explain the complex whole liver spectrum with only 4 % remaining residual random error.

As for the formal aspects, the manuscript is logical, well-written and readable, however, there are some typos that should be corrected in the final paper.

The findings are interesting and open novel perspectives to determine donor liver quality. Some details, however, need to be clarified:

1. About what percent of transplanted livers are rejected do to IRI? Percent would be informative in introduction.

2. The numbers in Figure 2a represent characteristic peaks in Raman spectrum where shifts are detected. Numbers 1168 and 1333 are in the wrong order. Can we attribute the shifted peaks to specific molecules? Describe in figure legend if known molecules.

3. Have the authors analyzed mass spectrometry data of flash frozen tissue samples to identify other abundant molecules in rat liver that could exhibit substantial resonance enhancement and would contribute to 3RMR at 441 excitation wavelength?

4. On Suppl. Figure S1. a-f) labeling is missing from the graph. On graph d) weight gain at the end of perfusion show individual dots as on other Figures.

6. PLOS authors have the option to publish the peer review history of their article (what does this mean?). If published, this will include your full peer review and any attached files.

Reviewer #1: No

Reviewer #2: No

---

## [Author Response · Author response to Decision Letter 0]

15 Sep 2021

Please see the attached file - Response to Reviewers.

---

## [Decision Letter · Decision Letter 1]

7 Oct 2021

Non-invasive Quantification of the Mitochondrial Redox State in Livers During Machine Perfusion.

PONE-D-21-06813R1

Dear Dr. Tessier,

We’re pleased to inform you that your manuscript has been judged scientifically suitable for publication and will be formally accepted for publication once it meets all outstanding technical requirements.

Kind regards,

Partha Mukhopadhyay, Ph.D.

Section Editor

PLOS ONE

Additional Editor Comments (optional):

Reviewers' comments:

Reviewer's Responses to Questions

**Comments to the Author**

1. If the authors have adequately addressed your comments raised in a previous round of review and you feel that this manuscript is now acceptable for publication, you may indicate that here to bypass the “Comments to the Author” section, enter your conflict of interest statement in the “Confidential to Editor” section, and submit your "Accept" recommendation.

Reviewer #1: All comments have been addressed

Reviewer #2: All comments have been addressed

2. Is the manuscript technically sound, and do the data support the conclusions?

Reviewer #1: (No Response)

Reviewer #2: Yes

3. Has the statistical analysis been performed appropriately and rigorously? 

Reviewer #1: (No Response)

Reviewer #2: Yes

4. Have the authors made all data underlying the findings in their manuscript fully available?

Reviewer #1: (No Response)

Reviewer #2: Yes

5. Is the manuscript presented in an intelligible fashion and written in standard English?

Reviewer #1: (No Response)

Reviewer #2: Yes

6. Review Comments to the Author

Reviewer #1: (No Response)

Reviewer #2: The authors have adequately answered my questions and corrected related sections in the MS. I have no further questions.

7. PLOS authors have the option to publish the peer review history of their article (what does this mean?). If published, this will include your full peer review and any attached files.

Reviewer #1: No

Reviewer #2: No

---

## [Editor Report · Acceptance letter]

18 Oct 2021

PONE-D-21-06813R1 

Non-invasive Quantification of the Mitochondrial Redox State in Livers During Machine Perfusion 

Dear Dr. Tessier:

I'm pleased to inform you that your manuscript has been deemed suitable for publication in PLOS ONE. Congratulations! Your manuscript is now with our production department. 

Kind regards, 

on behalf of

Dr. Partha Mukhopadhyay 

Section Editor

PLOS ONE